# Measurement invariance of the strengths and difficulties questionnaire across socioeconomic status and ethnicity from ages 3 to 17 years: A population cohort study

Umar Toseeb[1]*, Olakunle Oginni[2,3], Richard Rowe[4], Praveetha Patalay[5]

**1** Department of Education, University of York, York, United Kingdom, **2** Institute of Psychiatry, Psychology and Neuroscience, King's College London, London, United Kingdom, **3** Department of Mental Health, Obafemi Awolowo University, Ile-Ife, Nigeria, **4** Department of Psychology, University of Sheffield, Sheffield, United Kingdom, **5** Centre for Longitudinal Studies and MRC Unit for Lifelong Health and Ageing, University College London, London, United Kingdom

* umar.toseeb@york.ac.uk

**Data Availability Statement:** The study is a secondary analysis of existing data from the Millennium Cohort Study. Data can be accessed via

## Abstract

Mental health inequalities along ethnic and socioeconomic groupings are well documented. The extent to which these observed inequalities are genuine or reflect measurement differences is unclear. In the current study we sought to investigate this in a large population-based sample of children and adolescents in the United Kingdom. The main objective of the study was to establish whether the parent-report Strengths and Difficulties Questionnaire (SDQ) was invariant across ethnicity and socioeconomic status groupings at six time points from 3 to 17 years (maximum N = 17,274). First, we fitted a series of confirmatory factor analysis models to the data and confirmed that the five-factor structure (emotional problems; peer problems; conduct problems; hyperactivity/inattention; and prosocial behaviour) had acceptable fit at ages 5, 7, 11, and 14 years. Next, we tested configural, metric, and scalar invariance at these time points and demonstrated scalar invariance across household income, parent highest education, and ethnicity categories. The five-factor structure did not fit well at ages 3 and 17 years; therefore invariance was not tested at these ages. These findings suggest the parent-report SDQ can be used to measure socioeconomic and ethnic inequalities in mental health from ages 5–14 years but more consideration is required outside these ages.

## Introduction

Identifying mental health difficulties during childhood and adolescence is important for providing support to those affected. Whilst mean differences between socioeconomic and ethnic categories are well documented [e.g., 1], the challenge of consistently measuring mental health difficulties between groups remains; it is unclear whether existing measures assess the same constructs across groups comparably. This raises further questions about whether comparing

the the UK Data Service (http://www.ukdataservice.ac.uk) for researchers who meet the criteria for access to confidential data.

**Funding:** The author(s) received no specific funding for this work

**Competing interests:** The authors have declared that no competing interests exist.

group means appropriately assesses mental health inequalities rather than reflecting measurement artefacts.

Standardised psychopathology assessments may show systematic socioeconomic and ethnic biases. Many measures were developed in White populations and may not reflect symptom profiles common in ethnically diverse backgrounds [2]. Similarly, children and adolescents from privileged socioeconomic backgrounds may display different symptom profiles compared to those from less privileged backgrounds [3] or their parents might identify and report difficulties differently. To increase understanding of potential measurement differences, measurement invariance was tested across different levels of socioeconomic status and ethnic groups in a nationally representative sample children and adolescents in the United Kingdom (UK) at multiple ages from 3 to 17 years.

## The strengths and difficulties questionnaire

We focussed on the parent-report Strengths and Difficulties Questionnaire [SDQ; 4]; a widely used screening tool for psychopathology in children and adolescents. The SDQ contains 25 items forming five subscales each with five statements rated as 'Not true', 'Somewhat true' and 'Certainly true'. Four of the five subscales capture difficulties: emotional problems, peer problems, conduct problems, and hyperactivity/inattention. The fifth subscale captures strengths: prosocial behaviour.

The SDQ may be scored as: a single total difficulties score (summing the four problems subscales); two difficulties scores -internalising (summing emotional and peer problems) and externalising (summing conduct problems and hyperactivity/inattention); or four difficulties scores (emotional problems, peer problems, conduct problems, and hyperactivity/inattention). The prosocial behaviour subscale is usually scored separately. The four-subscale structure corresponds most closely to discrete symptoms of psychiatric conditions; emotional problems–depression and anxiety, conduct problems–conduct disorder, and hyperactivity/inattention–attention deficit hyperactivity disorder. The peer problems subscale identifies social isolation and peer related interpersonal difficulties rather than symptoms of discrete psychiatric conditions.

The factor structure of different SDQ scoring methods has been assessed extensively. These include a three-factor structure [5], a five-factor factor structure [6], and bi-factor and second order structures [7]. Direct comparisons of the various structures in the present study sample [8] and others [7, 9, 10], suggest the five-factor structure fits best and, therefore, was tested in the present study.

## Socioeconomic and ethnic inequalities in mental health difficulties

Socioeconomic and ethnic inequalities in child and adolescent psychopathology are frequently reported. Adolescents from South Asian or Black ethnic groups in the UK experience higher levels of mental health difficulties compared to their White counterparts [11]. Other studies report that ethnic minorities might have fewer mental health difficulties compared to their White peers [12, 13]. These reported ethnic inequalities might also be sensitive to developmental periods. In one study, ethnic minorities in the UK had more mental health difficulties in early childhood than their White counterparts but by early adolescence ethnic minorities had fewer difficulties [14]. Such reported inequalities are usually explained in two ways; a consequence of belonging to the ethnic group itself (e.g., cultural practices) and or as a result of social, educational, or economic disadvantage experienced by these groups [15]. Indeed, evidence suggests that some, but not all, of the reported ethnic inequalities in mental health difficulties can be explained by socioeconomic status [16].

Socioeconomic status itself is an independent predictor of child and adolescent psychopathology. The effects of socioeconomic status on mental health difficulties are likely to manifest through various pathways. For example, children from low socioeconomic backgrounds have less access to material and social resources to support social, emotional, and cognitive development, which is in turn linked to the development of psychopathology [17]. Children from low socioeconomic backgrounds are also more likely to experience trauma [18], which is associated with poor mental health [19]. Additionally, socioeconomic pressures may negatively affect parent-child relationships, which have a knock-on effect on development [20, 21]. There is, therefore, consistent evidence of a *measured* difference in mental health difficulties across socioeconomic and ethnic groups.

What remains unclear, however, is the extent to which measurement characteristics contribute to the observed differences. To illustrate, it may be that individuals from certain ethnic minorities or those from lower socioeconomic backgrounds have lower levels of reading ability or language proficiency [22] and so are less able to understand questionnaire items. Similarly, even if well understood, the instruments used to measure mental health difficulties may measure different constructs across groups. That is, even if the same items are being used to measure difficulties across groups, the likelihood of certain items being endorsed may depend on belonging to certain groups. For example, in one study, Black individuals were less likely to endorse questions on worthlessness and suicidal ideation compared to White individuals with the same level of depression [23]. Therefore, before comparing mental health difficulties between socioeconomic and ethnic groups, it is necessary to establish construct invariance.

## Construct invariance

Construct invariance (or construct equivalence) refers to whether a psychological construct has the same meaning between groups [24]. In this case, whether mental health difficulties as measured using the SDQ items hold the same meaning across different socioeconomic and ethnic groups. Construct invariance can be categorised and assessed at three hierarchical levels [25]. Configural invariance assesses whether factor structure is equivalent across groups, i.e., similarity of the number of factors and pattern of factor loadings across groups. Metric invariance assesses whether, in addition to configural invariance, the magnitude of factor loadings is equivalent across groups. Practically, the highest level of invariance is scalar invariance, which refers to whether, in addition to metric invariance, thresholds are equivalent between groups. This means that individuals in comparison groups who are on equivalent levels of the latent factor will have equivalent scores on the indicator variables i.e., the comparison groups are interpreting the responses to the indicator variables in the same way [26]. Therefore, achieving scalar invariance would imply that any cross-group differences in mean SDQ scores are due to genuine differences in mental health difficulties rather than group-related differences in how the measure is understood and completed.

## The current study

Longitudinal and gender invariance during childhood and adolescence were recently established for the parent-report SDQ using data from the Millennium Cohort Study [27], which will also be used in the present study. To our knowledge, SDQ measurement invariance across socioeconomic and ethnic categories across childhood and adolescence in the UK has not been fully investigated. In one study, researchers investigated measurement invariance between British Indian children and their White peers [28]. They found that the parent-, teacher-, and child-report SDQs were invariant for British Indian and White children. The focus on British Indian children, however, does not provide any indication of whether the

measure is suitable to assess inequalities across a range of other ethnic minority groups in the UK. This is important considering that there are many more minority ethnic groups in the UK of which British Indian families tend to be the least disadvantaged of the UK ethnic minorities on a range of indicators [29]. In another study, researchers investigated the factor structure of the child-report SDQ across seven different countries [30] but did not include the UK. In a further UK-based study, researchers tested the factor structure of the parent-report SDQ in a multi-ethnic UK cohort [31]. They were unable to demonstrate a consistent factor structure across ethnic groups, but their relatively small sample consisted of 3–5 year olds disproportionally representing one ethnic minority, Pakistani. To the best of our knowledge, no previous study has investigated invariance based on multiple indictors of socioeconomic status (i.e., income and education). We addressed these gaps in previous research.

We sought to assess construct invariance across socioeconomic and ethnic groups in a large UK representative sample at multiple ages through childhood and adolescence. We were motivated by two research questions. First, to what extent does the five-factor structure of the SDQ (emotional problems, peer problems, conduct problems, hyperactivity/inattention, prosocial behaviour) fit the parent-report data across early childhood and adolescence at ages 3, 5, 7, 11, 14 and 17 years? (research question 1). Based on previous research in this sample [32], we expected the five factor structure to have at least adequate fit at ages 5–14 years but not at ages 3 and 17 years. Second, to what extent is there socioeconomic- and ethnicity-based measurement invariance for the SDQ during early childhood and adolescence? (research question 2). Given the paucity of previous research assessing measurement invariance across these groups we had no hypothesis about whether conditions of invariance will be met.

## Materials and methods

### Ethical approval

The study was performed in accordance with the ethical standards as laid down in the 1964 Declaration of Helsinki and its later amendments. Ethical approval for data collection for Millennium Cohort Study [MCS, 33] was granted by the National Health Service Research Ethics Committee. Participants provided written consent. Full details of the ethical process for the MCS are available at https://cls.ucl.ac.uk/wp-content/uploads/2017/07/MCS-Ethical-Approval-and-Consent-2019.pdf.

### Study design

The study was a secondary analysis of existing data from a prospective cohort study. Our analysis was cross-sectional as we fitted models separately at each time point rather than including longitudinal relationships between variables.

### Participants

The MCS [33] is a multi-disciplinary longitudinal study following approximately 19,000 UK children born between 2000–2002. Full sampling details can be found elsewhere [34 and https://cls.ucl.ac.uk/cls-studies/millennium-cohort-study/]. Briefly, child benefit records, which was a universal social security benefit for families with at least one child, were screened to identify eligible families. Families first participated when their child was 9 months old and were followed-up when the child was 3, 5, 7, 11, 14 and 17 years of age. Researchers were trained to administer surveys and they conducted interviews in family homes at each wave of data collection.

We used data from 17,274 participants (49% female) in our analyses. A breakdown of the sample by socioeconomic and ethnicity indicators is provided in Table 1. Participants were included if parent-report SDQ data was available for at least one data collection point. Some families had more than one child taking part in the study. To avoid nesting effects only one child per family (selected at random) was included.

## SDQ

The parent-report SDQ [4] was completed by a selected caregiver (the mother for >95% of the sample) at each contact. As described previously, the SDQ is a 25-item questionnaire consisting of five subscales each with five statements. The caregiver indicated the extent to which each statement described their child over the last six months on a three-point scale (0 = *not true*, 1 = *somewhat true*, 2 = *certainly true*). Items contained in each subscale are listed in the supporting information.

## Ethnicity and socioeconomic status measures

**Household income.**   Primary caregivers reported income from all sources (government benefits, employment etc.) when the child was three years old. If age three data was missing, then responses collected when the child was nine months old were used. The Organisation for Economic Co-operation and Development modified scale was used to standardise overall household income [35]. This was then used to create quintiles (1 = *lowest income*, 5 = *highest income*).

**Parent highest education.**   Primary caregivers reported their highest education level when the child was three years old (or age nine months, if data were missing at age three). These were converted into five categories (1 = *O-Level / GCSE grades D-G*, 2 = *O-Level / GCSE grades*

**Table 1. Sample demographics.**

|  | Age 3 | Age 5 | Age 7 | Age 11 | Age 14 | Age 17 |
|---|---|---|---|---|---|---|
| **Household Income** | **14,810 (100%)** | **14,729 (100%)** | **13,447 (100%)** | **12,784 (100%)** | **11,230 (100%)** | **9,258 (100%)** |
| Lowest Quintile | 3,060 (21%) | 3,283 (22%) | 2,857 (21%) | 2,756 (22%) | 2,329 (21%) | 1,785 (19%) |
| 2nd Quintile | 3,186 (22%) | 3,208 (22%) | 2,858 (21%) | 2,711 (21%) | 2,383 (21%) | 1,895 (20%) |
| 3rd Quintile | 2,967 (20%) | 2,902 (20%) | 2,689 (20%) | 2,540 (20%) | 2,246 (20%) | 1,872 (20%) |
| 4th Quintile | 2,878 (19%) | 2,752 (19%) | 2,575 (19%) | 2,428 (19%) | 2,142 (19%) | 1,852 (20%) |
| Highest Quintile | 2,719 (18%) | 2,584 (18%) | 2,468 (18%) | 2,349 (18%) | 2,130 (19%) | 1,854 (20%) |
| **Parental Highest Education** | **11,665 (100%)** | **11,557 (100%)** | **10,625 (100%)** | **10,089 (100%)** | **8,825 (100%)** | **7,390 (100%)** |
| O Level/ GCSE Grade D-G | 1,523 (13%) | 1,485 (13%) | 1,320 (12%) | 1,265 (13%) | 1,073 (12%) | 815 (11%) |
| O Level/ GCSE Grade A*-C | 4,884 (42%) | 4,864 (42%) | 4,434 (41%) | 4,151 (41%) | 3,528 (40%) | 2,907 (40%) |
| A / AS / & Equivalent | 1,412 (12%) | 1,403 (12%) | 1,312 (12%) | 1,249 (12%) | 1,081 (12%) | 933 (13%) |
| Diploma in Higher Education & Equivalent | 1,339 (11%) | 1,310 (11%) | 1,190 (11%) | 1,134 (11%) | 1,031 (12%) | 863 (12%) |
| First degree or Higher | 2,507 (21%) | 2,495 (22%) | 2,369 (22%) | 2,290 (23%) | 2,112 (24%) | 1,872 (25%) |
| **Ethnicity** | **14,820 (100%)** | **14,762 (100%)** | **13,478 (100%)** | **12,814 (100%)** | **11,257 (100%)** | **9,281 (100%)** |
| White | 12,704 (86%) | 12,565 (85%) | 11,443 (85%) | 10,768 (84%) | 9,285 (82%) | 7,628 (78%) |
| Mixed | 432 (3%) | 428 (3%) | 375 (3%) | 374 (3%) | 317 (3%) | 273 (3%) |
| South Asian | 1,074 (7%) | 1,136 (7%) | 1,086 (8%) | 1,118 (8%) | 1,132 (10%) | 944 (10%) |
| Black | 404 (3%) | 424 (3%) | 385 (3%) | 363 (3%) | 326 (3%) | 270 (3%) |
| Other | 206 (1%) | 209 (1%) | 189 (1%) | 191 (1%) | 197 (2%) | 166 (2%) |

Values are N (%). % are a function of the superordinate categories in bold.

$A^*$-C, 3 = A / AS / & Equivalent, 4 = *Diploma in Higher Education & Equivalent*, 5 = *First degree or higher*). A higher category indicates a higher level of education.

**Ethnicity.** Primary caregivers reported their child's ethnicity. Initially, eight categories were derived using Office for National Statistics categories: White, Mixed, Black African, Black Caribbean, Asian Indian, Asian Pakistani, Asian Bangladeshi, and Other. The first set of models were fitted models using these ethnicity groupings. However, some models did not converge due to small participant numbers in some groups (e.g., there were only 91 Black Caribbean adolescents at age 17 years). Subsequently, we reduced these groupings to five categories by combining Black African and Black Caribbean to create a single Black category and Asian Indian, Asian Pakistani, and Asian Bangladeshi into a single South Asian category.

## Statistical analysis

Data cleaning, preparation, and reliability analyses were done in STATA/MP 17.0 [36]. Invariance models were fitted in Mplus 8.2 [37].

To address research question 1, confirmatory factor analysis models were fitted separately at age 3, 5, 7, 11, 14, and 17 years. We tested the commonly used five factor structure of the SDQ; each of the 25 items were loaded on to one of the five latent factors; emotional problems, peer problems, conduct problems, hyperactivity/inattention, and prosocial behaviour as has been previously described [32]. The fit indices of the confirmatory five-factor models were assessed to determine how well the models fit the data. Models with satisfactory fit indices (see below) were considered to fit the data well and deemed appropriate for further analyses. We did not test alternative factor structures as previous work in this sample demonstrated that the five-factor structure fitted the data better than other structures [32]. SDQ items were treated as ordinal variables using the weighted least-squares means and variances-adjusted estimator. Model fit was considered adequate where comparative fit index [CFI, 38] values were >0.90 and root mean square error of approximation [RMSEA, 39] values were <0.06 [40].

To address research question 2, measurement invariance was tested at ages demonstrating adequate model fit (i.e., ages 5, 7, 11, and 14 years). The aim of our analyses was not to test longitudinal measurement invariance; this has already been demonstrated in this sample [32] and others [27]. Instead, we were specifically concerned with between-group invariance at each of the time points separately. Doing this allowed us to determine the extent to which group comparisons along socioeconomic and ethnicity groupings reflect genuine group differences. Models were fitted using the Mplus command "model: configural metric scalar". This command fits three nested models. In the configural model, the five-factor structure was imposed without constraining factor loadings or thresholds. The metric model fixed factor loadings equal across all categories. The scalar model fixed both factor loadings and thresholds equal across categories. Non-invariance was assumed when there was a change in CFI of ≥ -.010 combined with either a change in RMSEA of ≥ .015 or a change in standard mean root residual (SRMR) of ≥ .030, as described in previous work [25]. At each age we separately tested invariance across household income, parental highest education, and ethnicity.

A sample and non-response weight [41] was used in the invariance analysis to account for sample attrition and oversampling of ethnic minority and low-income households.

## Missing data

There were missing data at each of the six time points. Relative to the original sample, response rates were 81% (age 3), 79% (age 5), 72% (age 7), 69% (age 11), 61% (age 14) and 74% (age 17). We did not fit longitudinal models and so do not report missingness at each time point as a function of the overall sample for the SDQ variables. There was missing data for ethnicity (7

observations, 0%), household income (51 observations, 0%), and parental highest education (4,245 observations, 25%), which was assumed to be missing at random. Missing highest education data was dependent on ethnicity and household income. Missing data was handled using the weighted least squares mean estimator in Mplus.

## Results

### Descriptive statistics

Both the overall and stratified means (by socioeconomic and ethnicity indicators) of the SDQ subscales at each timepoint are shown in S1-S5 Tables in S1 File. Whilst we did not test change over time, we describe general trends in means here. Regarding overall means, there was a general trend of emotional difficulties and peer problems increasing between ages 3 and 17 years. Conversely, conduct problems and hyperactivity/inattention decreased between the ages 3 and 17 years. Prosocial behaviour increased between ages 3 and 11 years after which there was a decrease.

For both socioeconomic indicators, household income and parental highest education, there was a clear pattern: each SDQ problem subscale (emotional difficulties, peer problems, conduct problems, hyperactivity/inattention) decreased with each quintile increase in household income or parent highest education across all timepoints. In contrast, prosocial behaviour increased with each quintile increase in household income or parent highest education at each of the six time points.

For ethnicity, there was no clear pattern. Generally, Black or White children had the fewest difficulties while Mixed, South Asian, and Other ethnicity children had the most difficulties; but this varied by age and the SDQ subscale.

### Internal consistency

The Cronbach's alphas for the SDQ subscales ranged from .47 to .71 at age 3 years, .52 to .77 at age 5 years, .58 to .79 at age 7 years, .63 to.79 at age 11 years, .62 to.76 at age 14 years, and .61 to .76 at age 17 years (S6 Table in S1 File). The Cronbach's alphas tended to increase with age. The peer problems subscale generally had the lowest Cronbach's alphas (except at 11 years) while the hyperactivity subscale had the highest Cronbach's alphas per timepoint.

The McDonald's omega values followed a similar pattern to the Cronbach's alphas (see S6 Table in S1 File), ranging from .47 to .72 at age 3 years, .52 to .77 at age 5 years, .59 to .79 at age 7 years, .66 to .79 at age 11 years, .64 to .78 at age 14 years, and .63 to .77 at age 17 years. The McDonald's omega values were also generally lowest for peer problems and highest for hyperactivity within each time point.

### Factor structure

The five-factor structure of the SDQ was tested using a series of CFAs; one at each age. The model fit statistics for these CFAs are shown in Table 2. The five-factor structure had adequate fit at ages 7, 11, and 14 years. All CFI values were >.90 and all RMSEA values were < .06. At age 5 years the CFA was .899. Given that the RMSEA was < .06, the CFI was very close to the threshold and consistent with the recommendation to consider multiple fit indices (Kline, 2016), the fit at age 5 years was also considered adequate. These results were consistent with previous research using the present and other samples [27, 32]. The five-factor structure did not fit well at ages 3 and 17 (CFIs < .90). Therefore, we focussed invariance analyses on ages 5–14.

**Table 2. Model fit statistics for confirmatory factor analyses.**

| Age (years) | N | CFI | RMSEA [95% CI] | SRMR |
|---|---|---|---|---|
| **3** | 14,824 | .853 | .056 [.055, .057] | .077 |
| **5** | 14,763 | .899 | .046 [.046, .047] | .066 |
| **7** | 13,479 | .917 | .044 [.043, .045] | .063 |
| **11** | 12,816 | .910 | .043 [.042, .043] | .058 |
| **14** | 11,258 | .929 | .036 [.035, .037] | .066 |
| **17** | 9,283 | .884 | .020 [.019, .021] | .069 |

N = sample size, CFI = comparative fit index, RMSEA = root mean square error of approximation, CI = confidence intervals, SRMR = standardised root mean squared.

## Measurement invariance

Table 3 shows that configural invariance was demonstrated across household income quintiles, with acceptable configural model fits at ages between 5–14 years. Metric invariance was achieved at each timepoint; constraining factor loadings to be equal across household income quintiles resulted in the CFIs, RMSEAs, and SRMRs either improving or decreasing within acceptable limits as described in the statistical analysis section. Similarly, scalar invariance was achieved at each age; further constraining thresholds across household income quintiles also resulted in CFIs, RMSEAs, and SRMRs, either improving or not decreasing beyond pre-specified acceptable limits.

Measurement invariance testing across parental education (Table 4) and ethnicity (Table 5) showed similar results; configural, metric and scalar invariance models demonstrated acceptable fit at all timepoints allowing scalar invariance to be accepted.

**Table 3. Household income measurement invariance models.**

| | CFI | ΔCFI | RMSEA [95% CI] | Δ RMSEA | SRMR | Δ SRMR |
|---|---|---|---|---|---|---|
| **Age 5 years (n = 14,729)** | | | | | | |
| Configural | .901 | - | .043 [.043, .044] | - | .072 | - |
| Metric | .906 | .005 | .041 [.040, .042] | -.002 | .073 | .001 |
| Scalar | .908 | .003 | .040 [.039, .040] | -.001 | .073 | .000 |
| **Age 7 years (n = 13,447)** | | | | | | |
| Configural | .916 | - | .042 [.041, .043] | - | .069 | - |
| Metric | .921 | .005 | .040 [.039, .041] | -.002 | .069 | .000 |
| Scalar | .925 | .004 | .038 [.037, .039] | -.002 | .070 | .001 |
| **Age 11 years (n = 12,784)** | | | | | | |
| Configural | .917 | - | .041 [.040,.042] | - | .066 | - |
| Metric | .921 | .004 | .039 [.038, .040] | -.002 | .067 | .001 |
| Scalar | .925 | .004 | .037 [.036, .038] | -.002 | .067 | .000 |
| **Age 14 years (n = 11,230)** | | | | | | |
| Configural | .921 | - | .038 [.037,.039] | - | .073 | - |
| Metric | .921 | .000 | .037 [.036, .038] | -.001 | .073 | .000 |
| Scalar | .925 | .004 | .035 [.034, .036] | -.002 | .074 | .001 |

N = sample size, CFI = comparative fit index, RMSEA = root mean square error of approximation, CI = confidence intervals, SRMR = standardised root mean squared, Δ = change

**Table 4. Parent highest education measurement invariance models.**

|  | CFI | ΔCFI | RMSEA [95% CI] | Δ RMSEA | SRMR | Δ SRMR |
|---|---|---|---|---|---|---|
| **Age 5 years (n = 9,062)** |  |  |  |  |  |  |
| Configural | .904 | - | .042 [.041, .043] | - | .074 | - |
| Metric | .912 | .004 | .039 [.038, .040] | -.002 | .075 | .001 |
| Scalar | .916 | .004 | .037 [.036, .038] | -.002 | .075 | .000 |
| **Age 7 years (n = 10,625)** |  |  |  |  |  |  |
| Configural | .929 | - | .040 [.038, .041] | - | .070 | - |
| Metric | .932 | .003 | .038 [.037, .039] | -.002 | .071 | .001 |
| Scalar | .933 | .001 | .036 [.035, .037] | -.002 | .071 | .000 |
| **Age 11 years (n = 7,799)** |  |  |  |  |  |  |
| Configural | .930 | - | .040 [.039, .041] | - | .068 | - |
| Metric | .935 | .005 | .038 [.036, .039] | -.002 | .068 | .000 |
| Scalar | .940 | .005 | .035 [.034, .037] | -.003 | .068 | .000 |
| **Age 14 years (n = 6,713)** |  |  |  |  |  |  |
| Configural | .931 | - | .035 [.033, .036] | - | .075 | - |
| Metric | .932 | .001 | .034 [.032,.035] | -.001 | .076 | .001 |
| Scalar | .936 | .004 | .032 [.032, .033] | -.002 | .076 | .000 |

N = sample size, CFI = comparative fit index, RMSEA = root mean square error of approximation, CI = confidence intervals, SRMR = standardised root mean squared, Δ = change

## Discussion

We tested socioeconomic- and ethnicity-based invariance of the parent-report SDQ from age 3 to 17 years in a large representative UK population sample. Achieving invariance across socioeconomic and ethnicity groupings is critical for making meaningful comparisons across

**Table 5. Ethnicity measurement invariance models.**

|  | CFI | ΔCFI | RMSEA [95% CI] | Δ RMSEA | SRMR | Δ SRMR |
|---|---|---|---|---|---|---|
| **Age 5 years (n = 14,762)** |  |  |  |  |  |  |
| Configural | .921 | - | .036 [.035, .037] | - | .070 | - |
| Metric | .925 | .004 | .034 [.033, .035] | -.002 | .071 | .001 |
| Scalar | .928 | .003 | .033 [.032, .033] | -.001 | .071 | .000 |
| **Age 7 years (n = 13,478)** |  |  |  |  |  |  |
| Configural | .933 | - | .035 [.034, .036] | - | .072 | - |
| Metric | .935 | .002 | .034 [.033, .035] | -.001 | .073 | .001 |
| Scalar | .938 | .003 | .032 [.031, .033] | -.002 | .073 | .001 |
| **Age 11 years (n = 12,814)** |  |  |  |  |  |  |
| Configural | .946 | - | .031 [.030, .032] | - | .066 | - |
| Metric | .949 | .003 | .030 [.029, .031] | -.001 | .067 | .001 |
| Scalar | .953 | .004 | .028 [.027, .029] | -.002 | .068 | .002 |
| **Age 14 years (n = 11,257)** |  |  |  |  |  |  |
| Configural | .944 | - | .026 [.025, .027] | - | .080 | - |
| Metric | .943 | -.001 | .025 [.024, .026] | -.001 | .080 | .000 |
| Scalar | .946 | .003 | .024 [.023, .025] | -.001 | .081 | .001 |

N = sample size, CFI = comparative fit index, RMSEA = root mean square error of approximation, CI = confidence intervals, SRMR = standardised root mean squared, Δ = change

these groups. Without this, such comparisons may represent measurement differences rather than genuine differences. The results of our study demonstrate scalar invariance for household income, parent highest education, and ethnicity categories for parent-report SDQ between ages 5 and 14 years. This suggests that the parent-report SDQ can be used to meaningfully compare inequalities in mental health across socioeconomic and ethnicity indicators from ages 5 to 14 years.

These findings have implications for SDQ users in multicultural settings like the UK. Many population-based studies, including the MCS, use the SDQ to screen for psychopathology in samples from multiple ethnic and socioeconomic backgrounds within the same geographical locations. The practical implications of our findings is that they provide confidence in previous work using parent-report SDQ to compare mental health difficulties across socioeconomic [e.g., 14] and multiple ethnicity groupings [e.g., 12]. In other words, the parent-reported SDQ is a valid instrument for comparing mental health difficulties in young people from different socioeconomic and ethnic backgrounds. Along with recent research demonstrating longitudinal invariance using the present sample [32], our findings suggest that the parent-report SDQ can be further used to investigate how socioeconomic- and ethnicity-based inequalities in mental health develop longitudinally during childhood and adolescence. Building on this, the SDQ can subsequently be used to map changes in mental health difficulties during development and how these differ across socioeconomic and ethnic groups using alternative approaches such as growth curve modelling. Such an approach could further justify the investigation of developmentally sensitive periods in emerging inequalities amongst different socioeconomic and ethnic groups using an invariant instrument such as the SDQ.

However, our findings do not support comparisons across socioeconomic and ethnic groupings outside the 5 to 14 years age range. Consistent with previous analyses in this sample [32], configural invariance was not achieved at ages 3 and 17 years, meaning that the parent-report SDQ data did not fit the five-factor structure. Theoretically, these findings might imply that in early childhood, symptoms of mental health difficulties manifest differently than during later childhood and adolescence; or that parent-report SDQ at this age has a different factor structure compared to later assessments. The lack of configural invariance for the five-structure factor at age 17 years seems intuitive. One possible explanation might be that by this age, adolescents spend more time away from their parents (e.g., with friends) and so parents might be less aware of the difficulties their child might be experiencing. Adolescents at this age are able to self-report symptoms of mental health difficulties, in contrast to early childhood. An important implication of the absence of socioeconomic and ethnic invariance for the parent-reported SDQ at ages 3 and 17 years is that further investigations are needed to understand whether alternative factor structures of the SDQ or other measures of psychopathology are invariant across socioeconomic and ethnic groupings outside the 5-14-year age group.

A number of strengths and weaknesses that should be considered when interpreting these findings. A major strength is the use of a large, longitudinal, and representative sample which incorporated participants from diverse backgrounds. Specifically, families from low socioeconomic backgrounds and ethnic minorities were oversampled, and this provided sufficient power for comparative analyses. Data from a prospective cohort study also allowed for testing invariance at multiple time points. Thus, while we did not test longitudinal invariance (this having been recently demonstrated in the present sample [32]), we demonstrated that socioeconomic and ethnic invariance were consistent at multiple timepoints during development. These findings may not, however, hold outside of the UK, and will have to be tested separately. Additionally, the ethnic groupings that we used in our study may not be ideal as there is a lot of heterogeneity within the commonly used ethnic groupings of White, South Asian, Black, Mixed, and Other in the UK. For example, in the UK, some studies have suggested that Indian

children tend to have a mental health advantage compared to Bangladeshi and Pakistani children [13]. Similarly, the Black grouping in our study might be problematic because it encompasses two culturally different groups (Black African and Black Caribbean), with substantial heterogeneity within these groups as well. Therefore, future research should consider these ethnic groups separately, where sample sizes allow. Our findings also need to be replicated both in the UK and other contexts in which there is a high socioeconomic and ethnic diversity.

## Conclusions

We demonstrate configural, metric and scalar invariance for parent-report SDQ between ages 5–14 years for household income, parent education, and ethnicity. Therefore, the parent-report SDQ measures comparable mental health constructs in children and adolescents from different socioeconomic and ethnic groups in the UK. Our findings validate previous use of parent-report SDQ to investigate mental health disparities among young people from different ethnic and socioeconomic backgrounds. Our findings also support the utility of the parent-report SDQ in the future assessment of mental health inequalities across socioeconomic and ethnic groupings in the UK, and how these change uniquely within different socioeconomic and ethnic groups.

## Supporting information

**S1 File. The supporting formation contains the SDQ items and the descriptive statistics and internal reliability measures for each SDQ subscale by age.**
(DOCX)

## Acknowledgments

We are grateful to the children and families who take part in the study. Data was accessed via the UK Data Service (http://www.ukdataservice.ac.uk/). The Centre for Longitudinal Studies, UCL Institute of Education, the UK Data Archive, and UK Data Service bear no responsibility for the analysis or interpretation of these data.

## Author Contributions

**Conceptualization:** Umar Toseeb, Olakunle Oginni, Richard Rowe, Praveetha Patalay.

**Formal analysis:** Umar Toseeb.

**Writing – original draft:** Umar Toseeb.

**Writing – review & editing:** Umar Toseeb, Olakunle Oginni, Richard Rowe, Praveetha Patalay.

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
