## [Decision Letter · Decision Letter 0]

12 Oct 2022

PONE-D-22-20945Measurement Invariance of the Strengths and Difficulties Questionnaire by Socioeconomic Status and Ethnicity from Ages 3 to 17 years: A Population Cohort StudyPLOS ONE

Dear Dr. Toseeb,

Thank you for submitting your manuscript to PLOS ONE. After careful consideration, we feel that it has merit but does not fully meet PLOS ONE’s publication criteria as it currently stands. Therefore, we invite you to submit a revised version of the manuscript that addresses the points raised during the review process.

An expert has reviewed your contribution and found some merits. However, he also pointed out some points that you should pay attention to prepare a revision. In addition to his comments, I have a major concern regarding your analysis in measurement invariance. Specifically, you have a longitudinal data but you did not test longitudinal invariance. I cannot understand this. If you used independent model to test the measurement invariance for the longitudinal dataset, your analysis did not consider the correlations nested in persons. Therefore, your invariance findings are biased. Please consider using dependent model to test for the longitudinal invariance. Also, this means that you have to consider the issue of missing values. 

We look forward to receiving your revised manuscript.

Kind regards,

Chung-Ying Lin

Academic Editor

PLOS ONE

Journal Requirements:

a) Did participants provide their written or verbal informed consent to participate in this study?

Reviewers' comments:

Reviewer's Responses to Questions

**Comments to the Author**

1. Is the manuscript technically sound, and do the data support the conclusions?

Reviewer #1: Yes

2. Has the statistical analysis been performed appropriately and rigorously? 

Reviewer #1: Yes

3. Have the authors made all data underlying the findings in their manuscript fully available?

Reviewer #1: No

4. Is the manuscript presented in an intelligible fashion and written in standard English?

Reviewer #1: Yes

5. Review Comments to the Author

Reviewer #1: Thank you very much for giving me the chance to review this manuscript.

I have a minor comment on the authors work before the manuscript got publishing as follows;

- Your study is focused on the invariance of the Strengths and Difficulties Questionnaire by Socioeconomic Status and Ethnicity for Mental Health Difficulties, which is unclear in the title. Therefore, could you modify your study title accordingly to your study objectives, the primary outcome, the study target population, and the study design?

- In the Abstract, please clearly highlight your main study objective.

-In the background of your study, could you highlight and discuss what has been done in the previous studies in terms of the factor structure and construct invariance of different SDQs across socioeconomic and ethnic categories and other factors investigated by these studies, including the results and the limitations in the methods you have mentioned.

- Under the methods section, please add a subsection for study design.

- Under the statistical analysis in the method, you mention that to address the research question, you performed confirmatory factor analysis models and measurement invariance; please clearly explain the steps of performing these analyses. Furthermore, in this section, you can refer to the indicators references with the references not to the results of your study regarding these indicators.

- The discussion is concise compared to your study results; in addition to discussing your study results, methods with previous studies are either inconsistent or similar; please highlight the theoretical and practical implications of your study findings.

- Moreover, at the end of the discussion, you have highlighted your study strengths and limitations, could you include the recommendations for future studies in line with the same points.

- In the conclusions, the most needing is a reflection of the results and the theoretical and practical implications.

6. PLOS authors have the option to publish the peer review history of their article (what does this mean?). If published, this will include your full peer review and any attached files.

Reviewer #1: **Yes: **Musheer A. Aljaberi

---

## [Author Response · Author response to Decision Letter 0]

7 Nov 2022

Response to Reviewers' Comments

PONE-D-22-20945- Title: Measurement Invariance of the Strengths and Difficulties Questionnaire Across Socioeconomic Status and Ethnicity from Ages 3 to 17 years: A Population Cohort Study

Editor Comments:

I have a major concern regarding your analysis in measurement invariance. Specifically, you have a longitudinal data but you did not test longitudinal invariance. I cannot understand this. If you used independent model to test the measurement invariance for the longitudinal dataset, your analysis did not consider the correlations nested in persons. Therefore, your invariance findings are biased. Please consider using dependent model to test for the longitudinal invariance. Also, this means that you have to consider the issue of missing values. We appreciate your comment as it has highlighted that we were not sufficiently clear on our aims in our original manuscript. The longitudinal measurement invariance of the parent reported SDQ has already been demonstrated in the MCS dataset that we are using (see Murray et al., 2022 in Assessment). Therefore, we do not believe further testing of longitudinal measurement invariance is required. Our contribution is to test invariance across ethnicity and socioeconomic status cross-sectionally at each of the time points separately. In doing this, researchers who want to test group-differences at any one of the time-points can meaningfully compare groups. Therefore within-person nesting did not confound our findings. We have made this clearer in the manuscript; specifically in “the current study” section of the introduction and also the “statistical analysis” section of the materials and method.

Reviewer #1: 

Thank you very much for giving me the chance to review this manuscript.

I have a minor comment on the authors work before the manuscript got publishing as follows;

- Your study is focused on the invariance of the Strengths and Difficulties Questionnaire by Socioeconomic Status and Ethnicity for Mental Health Difficulties, which is unclear in the title. Therefore, could you modify your study title accordingly to your study objectives, the primary outcome, the study target population, and the study design? We understand the reviewers concern here. However, we aim our paper at readers who are already familiar with the SDQ; as such they will be aware that it is a measure of mental health difficulties. In our opinion, adding “mental health difficulties” will over-complicate an already long title. However, we would be happy to reconsider this decision if the editor believes additional information in the title would be of value to the readership. 

- In the Abstract, please clearly highlight your main study objective. We have now made this clear in the abstract as “to establish whether the parent-report strengths and difficulties questionnaire (SDQ) is invariant across ethnicity and socioeconomic status groupings at six ages from 3 to 17 years (maximum N=17,274)”. We have also re-written the abstract to make our study and findings clearer.

-In the background of your study, could you highlight and discuss what has been done in the previous studies in terms of the factor structure and construct invariance of different SDQs across socioeconomic and ethnic categories and other factors investigated by these studies, including the results and the limitations in the methods you have mentioned. We have now included further details of this in “the current study” section of the introduction.

- Under the methods section, please add a subsection for study design. We have now added a sub-section for study design: “The study was a secondary analysis of existing data from a prospective cohort study.”

- Under the statistical analysis in the method, you mention that to address the research question, you performed confirmatory factor analysis models and measurement invariance; please clearly explain the steps of performing these analyses. We have now provided further information about the confirmatory factor analysis to provide more context for the interpretation of the steps undertaken for the measurement invariance analysis.

-Furthermore, in this section, you can refer to the indicators references with the references not to the results of your study regarding these indicators. We have now provided a reference to justify our preference for selectively investigating the five-factor model. We have also now stated in the Results section that our findings from the confirmatory factor analyses to those from previous research using the present and other samples.

- The discussion is concise compared to your study results; in addition to discussing your study results, methods with previous studies are either inconsistent or similar; please highlight the theoretical and practical implications of your study findings. The practical and research implications of our findings in multicultural settings like the UK have now been emphasised in the ‘Discussion’ section.

- Moreover, at the end of the discussion, you have highlighted your study strengths and limitations, could you include the recommendations for future studies in line with the same points. We have included a statement about suggestions for future research based on the limitation regarding our groupings of ethnic minorities. 

- In the conclusions, the most needing is a reflection of the results and the theoretical and practical implications. We have now summarised the implications of our findings for interpreting previous research and for informing future research.

---

## [Editor Report · Decision Letter 1]

16 Nov 2022

Measurement Invariance of the Strengths and Difficulties Questionnaire Across Socioeconomic Status and Ethnicity from Ages 3 to 17 years: A Population Cohort Study

PONE-D-22-20945R1

Dear Dr. Toseeb,

We’re pleased to inform you that your manuscript has been judged scientifically suitable for publication and will be formally accepted for publication once it meets all outstanding technical requirements.

Kind regards,

Chung-Ying Lin

Academic Editor

PLOS ONE

Additional Editor Comments (optional):

The authors have satisfactorily addressed the reviewer's comments and my concerns. Thank you for disclosing the information regarding the prior study on longitudinal invariance. For the revision, I only have one minor comment but I think it can be addressed during the proofread. That is, the sentence "Along with recent research demonstrating longitudinal invariance using the present (MCS), ...." does not need to have the parentheses for MCS. 
---

## [Editor Report · Acceptance letter]

19 Dec 2022

PONE-D-22-20945R1 

Measurement Invariance of the Strengths and Difficulties Questionnaire Across Socioeconomic Status and Ethnicity from Ages 3 to 17 years: A Population Cohort Study 

Dear Dr. Toseeb:

I'm pleased to inform you that your manuscript has been deemed suitable for publication in PLOS ONE. Congratulations! Your manuscript is now with our production department. 

Kind regards, 

on behalf of

Dr. Chung-Ying Lin 

Academic Editor

PLOS ONE